# Effects of Oral Alpha-Lipoic Acid Treatment on Diabetic Polyneuropathy: A Meta-Analysis and Systematic Review

**DOI:** 10.3390/nu15163634

**Published:** 2023-08-18

**Authors:** Ruey-Yu Hsieh, I-Chen Huang, Chiehfeng Chen, Jia-Ying Sung

**Affiliations:** 1Department of Neurology, Taipei Municipal Wan Fang Hospital, Taipei Medical University, Taipei 116, Taiwan; 2Division of Plastic Surgery, Department of Surgery, Wan Fang Hospital, Taipei Medical University, Taipei 116, Taiwan; 3Department of Public Health, School of Medicine, College of Medicine, Taipei Medical University, Taipei 110, Taiwan; 4Cochrane Taiwan, Taipei Medical University, Taipei 110, Taiwan; 5Department of Neurology, School of Medicine, College of Medicine, Taipei Medical University, Taipei 110, Taiwan

**Keywords:** alpha-lipoic acid, diabetes polyneuropathy, DSPN

## Abstract

Alpha-lipoic acid (ALA) was found to improve the symptoms in patients with diabetic sensorimotor peripheral neuropathy (DSPN) by reducing oxidative stress and ameliorating microcirculation. Our meta-analysis is aimed at evaluating the effects of oral-administered ALA versus a placebo in patients with DSPN and determining the optimal dosage for this treatment. We systematically reviewed randomized controlled trials (RCTs) in the PubMed, Embase, and Cochrane databases to determine the efficacy of oral ALA for patients with DSPN. The primary outcome was total symptoms’ score (TSS), and secondary outcomes were the neurological disability score (NDS), neuropathy impaired score (NIS), NIS-lower limb (NIS-LL), vibration perception threshold (VPT), nerve conduction study (NCS) results, and global satisfaction. A subgroup analysis of the ALA dosage (600, 1200, and 1800 mg/day) was also conducted. Ten RCTs (1242 patients) were included. ALA treatment produced favorable results for TSS (a dose-related trend was observed), NDS, and the global satisfaction score. For VAS, VPT, NIS-LL, and NCS results, ALA did not produce favorable results. ALA treatment had favorable effects on DSPN by reducing sensory symptoms, and it resulted in a dose-dependent response relative to the placebo for TSS and the global satisfaction score. The use of ALA to prevent neurological symptoms should be further researched.

## 1. Introduction

Diabetic neuropathies include heterogeneous presentations such as symmetric sensorimotor neuropathy, autonomic neuropathy, mononeuropathy, mononeuritis multiplex, polyradiculopathy, and plexopathy [1]. Among these neuropathies, symmetric sensorimotor neuropathy, which is also referred to as diabetic sensorimotor peripheral neuropathy (DSPN), is the most studied neuropathy.

DSPN is estimated to occur in 29% of patients with type 1 diabetes and 35% of those with type 2 diabetes in Asia [2] and in approximately 30% of the global population with diabetes [3]. Symmetric sensorimotor neuropathy, a length-dependent condition, can cause numbness, paresthesia, neuropathic pain, and even a severe sensory loss, affecting the hands and feet in a gloves-and-stocking pattern. Furthermore, cramping, weakness, and sensory ataxia develop in the later stage of symmetric sensorimotor neuropathy, causing not only discomfort but also disability [1]. Several risk factors influence the progression of diabetic neuropathy, including the duration of diabetes, poor glycemic control, and obesity [3,4].

Understanding the anatomy of the peripheral nervous system can provide insights into the mechanisms of DSPN. In the peripheral nervous system, only a few arterioles penetrate the endoneurium to supply the nerve fibers. When the blood flow cannot compensate the decrease in circulation, the individual’s peripheral nerves would be damaged to ischemia [5]. In addition, studies from Malik et al. also demonstrated that nerve fibers had ischemic changes secondary to a reduced endoneurial capillary density [6].

Hyperglycemia, mitochondrial dysfunction, inflammation, cell injury, and oxidative stress can also contribute to the progression of DSPN. The mechanism of diabetic neuropathy is not fully understood despite the research on this topic, and a disease-modifying treatment for this condition is yet to be developed.

Alpha-lipoic acid (ALA) is a caprylic acid-derived antioxidant that is synthesized in the mitochondria. Studies reported that ALA improved nitric-oxide-mediated endothelium-dependent vasodilation in patients with diabetes and improved microcirculation in patients with diabetic polyneuropathy [7]. An animal model study conducted in 1999 suggested that ALA is efficacious for moderate-ischemia–reperfusion injury, especially when the distal sensory nerves are affected [8]. Researchers have extensively studied the neuroprotective effects of ALA, which are achieved through reducing oxidative stress and increasing microcirculation [9]. However, the results pertaining to the optimal administration method and dosage have been inconclusive.

In 2004, Ziegler et al. conducted a meta-analysis of four trials, namely ALADIN I, ALADIN III, SYDNEY, and NATHAN II, to determine the efficacy and safety of treatments involving the intravenous administration of 600 mg of ALA over 3 weeks. Their results indicated that the treatment was safe and significantly alleviated positive neuropathic symptoms [10].

In 2012, a team of researchers performed a meta-analysis that evaluated the safety and efficacy of treatments involving the intravenous administration of ALA at a daily dose of 300–600 mg for 2–4 weeks. The results revealed significant improvements in positive neuropathic symptoms and the NCS result [11]. The research team subsequently conducted another meta-analysis to compare the effects of methylcobalamin (MC)-alone treatments versus MC + ALA treatments in patients with diabetic peripheral neuropathy; their results indicated that, relative to MC-only treatments, the daily administration of ALA (300–600 mg, intravenous administration) plus MC (500–1000 mg, intravenous or intramuscular administration) for 2–4 weeks was associated with a more favorable outcome in the nerve conduction study without severe adverse events [12]. A systematic review in 2022 conducted by Abubaker et al. [13] revealed that the use of ALA alone did not significantly improve neuropathic pain in patients with diabetes but still played a role in reducing neuropathic symptoms.

All the aforementioned meta-analyses, however, have focused on the treatment effects of intravenous ALA administration. By contrast, no high-quality study has examined the oral administration of ALA. In the present meta-analysis, we conducted a systematic search in accordance with the Preferred Reporting Items for Systematic Reviews and Meta-Analyses (PRISMA) guidelines. Our aim was to evaluate the effects of the oral administration of ALA versus a placebo in patients with DSPN and to identify the optimal dosage for this treatment.

## 2. Materials and Methods

The present study was performed in accordance with the Cochrane Handbook and PRISMA guidelines (Registration: INPLASY202340109).

### 2.1. Search Strategy

The PubMed, Embase, and Cochrane databases were searched for randomized controlled trials (RCTs) that evaluated the efficacy of ALA administration in patients with DSPN. The following search strategy and terms were used: (“polyneuropathies” [MeSH Terms] OR “polyneuropathies” [All Fields] OR “polyneuropathy” [All Fields]) AND (“diabetes” [All Fields] OR “diabetes mellitus” [MeSH Terms] OR (“diabetes” [All Fields] AND “mellitus” [All Fields]) OR “diabetes mellitus” [All Fields] OR “diabetes” [All Fields] OR “diabetes insipidus” [MeSH Terms] OR (“diabetes” [All Fields] AND “insipidus” [All Fields]) OR “diabetes insipidus” [All Fields] OR “diabetic” [All Fields] OR “diabetics” [All Fields] OR “diabetes” [All Fields]) AND (“thioctic acid” [MeSH Terms] OR (“thioctic” [All Fields] AND “acid” [All Fields]) OR “thioctic acid” [All Fields] OR (“lipoic” [All Fields] AND “acid” [All Fields]) OR “lipoic acid” [All Fields]). The literature search was conducted on 14 July 2022, and the results were updated on 12 September 2022. Only human participant studies in English were included. Furthermore, we manually checked the reference lists of the included studies to identify potentially eligible studies that could have been missed during the initial search.

### 2.2. Inclusion and Exclusion Criteria

Trials were included in the present meta-analysis if they (1) had an RCT design, (2) included adult patients with a primary diagnosis of diabetic sensorimotor polyneuropathy, (3) included patients with exposure to oral ALA, and (4) considered the following outcomes: total symptoms’ score (TSS), neuropathy impaired score (NIS), NIS-lower limb (NIS-LL), neurological disability score (NDS), visual analog scale of pain (VAS), vibration perception threshold (VPT), nerve conduction study (NCS) results, and global satisfaction score. Trials were excluded if they (1) did not collect any data of interest; (2) involved the intravenous administration of ALA; (3) included patients taking other supplements (e.g., MC or γ-linolenic acid) simultaneously; (4) were presented as abstracts, reviews, letters, or case reports; or (5) were not conducted as RCTs.

### 2.3. Neuropathy Assessment

#### 2.3.1. Primary Outcome: TSS

The TSS is calculated by summing the scores for the presence, severity, and frequency of four sensory neuropathic symptoms, namely lancinating pain, burning sensation, prickling sensation, and numbness during sleep. A TSS score can range from 0 to 14.64 [14] (Table 1).

#### 2.3.2. Secondary Outcome: NDS

The NDS includes the assessment scores for ankle reflex, vibration, pinprick, and temperature sensation on both sides of the great toes. An NDS score ranges from 0 to 10, with an NDS score of ≥6 indicating an abnormal status [15] (Table 2).

#### 2.3.3. Secondary Outcome: NIS and NIS-LL

The NIS is a composite score that reflects the severity of clinical impairments (weakness, reflex loss, and sensory loss), and it ranges from 0 to 244. To measure muscle weakness, 24 muscle groups are assessed, namely the cranial muscle groups (5 muscle groups), the upper body muscle groups (11 muscle groups), and the lower body muscle groups (8 muscle groups). The grading for weakness ranges from 0 (normal) to 1 (25% weak), 2 (50% weak), 3 (75% weak), 3.25 (able to move against gravity), 3.50 (movement, gravity eliminated), 3.75 (muscle contraction can be felt but no visible movement can be observed), and 4 (paralysis). To measure reflex loss, the biceps, triceps, brachioradialis, quadriceps, and ankle reflexes are graded as 0 (normal), 1 (decreased), or 2 (absent). Touch pressure, vibration, joint position, and pinprick are tested on the index fingers and great toes by using the aforementioned grading scale with endpoints ranging from 0 to 2 [16].

The NIS-LL is a derivative of the NIS that was designed to assesses the function of the lower limbs. The components assessed are the sensation (touch pressure, pinprick, vibration, and joint position), reflexes (quadriceps and triceps surae), and muscle weakness (hip flexion, hip extension, knee flexion, knee extension, ankle dorsiflexion, ankle plantar flexion, toe extension, and toe flexion) of the lower limbs. Each sensation or reflex item is scored as 0 (normal), 1 (decreased), or 2 (absent). Each muscle weakness item is scored as 0 (normal), 1 (25% weak), 2 (50% weak), 3 (75% weak), 3.25 (able to move against gravity), 3.5 (movement, gravity eliminated), 3.75 (muscle flickering with no observable movement), or 4 (paralysis). The maximum possible NIS-LL score is 88 [17].

#### 2.3.4. Secondary Outcome: VPT

The VPT is commonly tested using a 128-Hz tuning fork or neurothesiometer on the tips of the great toes. In the Garcia-Alcala et al. study [18], which is included in our meta-analysis, VPT was tested using a 128-Hz tuning fork applied bilaterally on the tip of the great toe. Responses were categorized as abnormal (no perception of vibration), present (examiner perceives vibration < 10 s after patient reported the disappearance of vibration perception), and reduced (examiner perceives vibration > 10 s after patient reported the disappearance of vibration perception). In El-Nahas et al.’s study [19], VPT was obtained with a neurothesiometer. In Zeigler et al.’s study in 2011 [20], it was not mentioned how they obtained the VPT outcome.

#### 2.3.5. Secondary Outcome: NCS

Studies that conducted NCSs differ in terms of the outcomes considered. In the present meta-analysis, sural sensory nerve action potential (SNAP), peroneal motor nerve conduction velocity (MNCV); median MNCV and SNAP; sensory nerve distal latency (SNDL); and ulnar MNCV, SNAP, and SNDL were included.

#### 2.3.6. Secondary Outcome: Global Satisfaction Score

The global satisfaction score reflects a patient’s overall satisfaction with an intervention, and it can be classified as very good, good, satisfactory, or unsatisfactory [20]. Because most of the studies included in the present meta-analysis consolidated good and very good scores into a single category, we also considered this category for this secondary outcome.

### 2.4. Data Extraction and Quality Assessment

Two authors (R.-Y.H. and I.-C.H.) independently extracted the following data: the first author, year of publication, country, number of patients in the ALA and placebo groups, dose of ALA administered, duration of therapy, and changes from baseline in TSS, NIS, NIS-LL, NDS, VAS, VPT, NCS, and the global satisfaction score. A standardized Microsoft Excel (16.75.2 version) file (Microsoft, Redmond, WA, USA) was used for data extraction. All disagreements between the authors (C.C. and J.-Y.S.) were resolved through discussions.

The methodological quality of each study was assessed using the Risk of Bias 2 tool, which was introduced in the Cochrane Handbook for Systematic Reviews of Interventions. The tool assesses five domains, including bias arising from the randomization process, bias caused by deviations from the intended interventions, bias caused by missing outcome data, bias in the measurement of the outcome, and bias in the selection of the reported results [21]. These domains were evaluated for all the included RCTs. All disagreements between the two aforementioned authors with respect to the bias assessment were resolved by reaching a final consensus among all the authors of the present meta-analysis. A traffic light plot was generated using the robvis tool 22 August 2019 version [22].

### 2.5. Statistical Analysis

The changes from baseline in TSS, NIS, NIS-LL, NDS, VPT, and NCS were treated as continuous outcomes; thus, all scores are expressed as mean differences (MDs) and standard deviations (SDs) with 95% confidence intervals (CIs). The heterogeneity among the studies was tested using the Cochrane Q χ2 test and *I*^2^ statistic. Studies with an *I*^2^ value of >50% or *p* of <0.1 were regarded as exhibiting heterogeneity. We employed a fixed-effect model to pool the estimates based on the presence or absence of heterogeneity. When considerable heterogeneity was present, we performed a sensitivity analysis to identify the possible reasons for the heterogeneity. When a given study did not provide change-from-baseline data, we used subsequently obtained data to assess the efficacy of the intervention. The global satisfaction score was determined based on the number of patients who graded a treatment as being very good or good. This outcome was treated as a noncontinuous outcome and is expressed as an odds ratio. The pooled SD from two groups was calculated using Cohen’s d in the DeCoMA tool 1.1 version [23].

A subgroup analysis of ALA dosages (600, 1200, and 1800 mg/day) was conducted to determine whether different doses of ALA had different effects relative to the placebo. A *p* value of <0.05 was regarded as statistically significant unless otherwise specified. All analyses were performed using the software Review Manager, version 5.4.1 (Nordic Cochrane Centre, Cochrane Collaboration, Copenhagen, Denmark).

## 3. Results

After an initial database search was performed, 512 (26 PubMed, 25 Embase, and 461 Cochrane studies) studies were identified. Among these studies, only 50 (14 PubMed, 13 Embase, and 23 Cochrane studies) studies were retained after title screening, and another 17 were subsequently excluded because they were duplicate studies. We conducted a full-text evaluation of the remaining 33 studies and excluded those that did not report any outcome of interest, used the same study population, adopted a non-RCT design, were not in English, or involved only intravenous ALA administration. In total, 10 RCTs met all our inclusion criteria and were included in the present meta-analysis (Figure 1).

The 10 included RCTs (published between 1999 and 2021) and the demographic characteristics of the participants in these RCTs are listed in Table 3. These RCTs had sample sizes ranging from 20 to 454 (a total of 1242 patients), and they were conducted in regions such as Europe (one study in Russia; three studies in Germany), North America (two studies in Mexico; one study in the United States), Asia (one study in India; one study in Pakistan), and Africa (one study in Egypt). The dosage of ALA varied across the studies, with seven RCTs prescribing 600 mg/day, four prescribing 1200 mg/day, and three prescribing 1800 mg/day. The mean ages of the participants ranged from 46.88 to 61.3 years, and their duration of diabetes mellitus ranged between 10.13 and 14.55 years. Several studies did not provide data pertaining to A1c levels, whereas the rest reported A1c levels ranging from 7.4% to 8.85%. The percentage of participants who were undergoing insulin treatment in each study mostly ranged between 43% and 57%, with an outlier of 96% being reported in one study [24]. Notably, none of the outcomes considered in the present meta-analysis were reported by more than 7 of the 10 included studies. Thus, we did not conduct publication bias assessment.

We assessed the quality of the included studies and presented the results as a traffic light plot in Figure 2. Two of the included studies were open-label RCTs, that is, information regarding the assigned treatment was not withheld from the trial participants or investigators [25,27]. A concern was identified with respect to the RCTs conducted by Tang et al. [28] and Ziegler et al. (a 2006 study [29]). Although both these studies were derived from the SYDNEY II trial, the outcomes reported by Tang et al. [28] were not reported by Ziegler et al. (a 2006 study [29]). Another concern was identified regarding the RCT conducted by Siddique et al. Specifically, they compared the pretreatment and post-treatment results for HbA1c, TSS, numbness sensation, burning sensation, and paresthesia instead of comparing the results between the treatment group and control group. Despite the aforementioned concerns, the studies were generally of high quality and exhibited a low risk of bias.

### 3.1. TSS

Six of the included RCTs reported TSS outcomes [14,18,20,24,27,29]. The pooled estimated effect, which was determined using a fixed-effect model, revealed that ALA administration led to significantly more favorable TSS outcomes relative to the control (MD, −1.69; 95% CI [−1.57, −1.08]). The shortest administered duration of the studies is 3 weeks and the longest is for 104 weeks. In this outcome, there are five studies that administered ALA at 600 mg/day, one study administered 1200 mg/day, and two studies administered 1800 mg/day. The heterogeneity of the studies was significant (*p* < 0.01); however, the reliability of this finding is low because only six RCTs were included. Furthermore, ALA administration produced favorable effects that exhibited dose-related trends (Figure 3).

### 3.2. NDS

Two of the included RCTs reported NDS outcomes [19,24]. The pooled estimated effect, which was determined using a fixed-effect model, revealed that ALA administration produced significantly more favorable NDS outcomes relative to the control (MD, −0.98; 95% CI [−1.29, −0.67]; Figure 4). The administered durations are 3 weeks and 24 weeks. In this outcome, there is one study that administered ALA at 1200 mg/day, and one study administered 1800 mg/day.

### 3.3. NIS

Three of the included RCTs reported NIS outcomes [14,20,29]. The pooled estimated effect, which was determined using a random-effect model, revealed that ALA administration produced significantly more favorable NIS outcomes relative to the placebo (MD, −1.16; 95% CI [−1.92, −0.41]; Figure 5). The shortest administered duration of the studies is 5 weeks and the longest is for 104 weeks. In this outcome, there are three studies that administered ALA at 600 mg/day, one study administered 1200 mg/day, and one study administered 1800 mg/day.

### 3.4. Global Satisfaction

Four of the included RCTs reported on the effects of ALA administration and the placebo on global satisfaction (i.e., good or very good ratings) [14,20,28,29]. The pooled odds ratio of ALA administration at the dosages of 600, 1200, and 1800 mg/day was 2.15 (95% CI [1.58, 2.92]), 3.2 (95% CI [1.33, 7.71]), and 6.56 (95% CI [2.60, 16.54]), respectively. The odds ratio for overall global satisfaction was 2.48 (95% CI [1.88, 3.27]) relative to the placebo. Furthermore, ALA administration produced favorable effects that exhibited dose-related trends (Figure 6). The shortest administered duration of the studies is 5 weeks and the longest is for 104 weeks. In this outcome, there are four studies that administered ALA at 600 mg/day, two studies administered 1200 mg/day, and two studies administered 1800 mg/day.

### 3.5. Parameters for Which Nonsignificant Results Were Reported

We also analyzed other parameters such as VAS, VPT, and NIS-LL. However, no significant results were reported for these parameters (VAS, MD [−0.32], 95% CI [−0.82, 0.19]; VPT, MD [−2.51], 95% CI [−7.40, 2.38]; NIS-LL, MD [−0.58], 95% CI [−1.27, 0.1]). Only a small number of RCTs reported NCS outcomes. To explore lower-limb-related outcomes, we analyzed peroneal MNCV and sural SNAP outcomes but did not identify any significant favorable outcome for these parameters resulting from the administration of ALA or the placebo (peroneal MNCV: MD, −0.13 and 95% CI [−0.82, 0.61]; sural SNAP: MD, 0.07 and 95% CI [−0.32, 0.46]). We analyzed median and ulnar nerve MNCV, SNAP, and SNDL outcomes but did not identify any significant results for these parameters.

## 4. Discussion

Reactive oxygen species (ROS) and reactive nitrogen species are byproducts of normal cellular metabolism and are produced with processes including NADPH-oxidase, myeloperoxidase, and nitric oxide synthase. Initially, reactive oxygen and nitrogen species (RONS) in the extracellular space form a part of the innate immune system that kills bacteria. However, the excessive release of RONS may cause damage to a host. An increasing number of studies and reviews have demonstrated that RONS are not only the byproducts of normal cellular metabolism but are also associated with the signaling for vascular tone, the synthesis of insulin, the activation of hypoxia-inducible factors, and the proliferation, differentiation, and migration of cells [30]. Furthermore, RONS trigger oxidative stress as a signaling messenger throughout the cell death pathways (apoptosis, necrosis, and autophagy) [31]. Additionally, free radicals can lead to the formation of another secondary radical, which may lead to oxidative stress and toxicity [32]. Thus, maintaining redox homeostasis is crucial.

Redox homeostasis is maintained with an endogenous defense system that involves enzymes such as superoxide dismutase, catalase, glutathione peroxidase, ascorbate, glutathione, flavonoids, tocopherol, carotenoid, and ubiquinol [31]. When ROS accumulate, endogenous molecules become insufficient for counteracting ROS, resulting in an increase in oxidative stress [33]. Oxidative stress has been reported to cause painful neuropathies, such as diabetic neuropathy, chemotherapy-induced neuropathy, peripheral-nerve-injury-induced neuropathic pain, and even poststroke neuropathic pain [33].

The activation of five pathways is a mechanism that can result in damage to the peripheral nerves, namely the polyol pathway (glucose metabolism), the accumulation of end-products of advanced glycosylation, the involvement of poly(ADP-ribose) polymerase, the hexosamine pathway, and the protein kinase C pathway. All these pathways are activated when the level of glucose is high; they can lead to vascular insufficiency and oxidative stress, which subsequently lead to nerve damage [34].

As a compound with antioxidant potential, ALA has been extensively studied as a potential treatment for neuropathic pain. However, limited evidence is available on how orally administered ALA affects diabetic polyneuropathy. In 2012, Mijnhout conducted a meta-analysis of four studies (653 patients) and revealed that the intravenous administration of ALA at a dosage of 600 mg/day over 3 weeks resulted in a significantly reduced TSS score; however, the meta-analysis did not explore the effects of orally administered ALA [35]. A study conducted in 2013 enrolled 1106 patients to compare the efficacy of a treatment combining lipoic acid (300–600 mg, intravenous administration) with MC (500–1000 mg, intravenous or intramuscular administration) against that of MC-alone treatment for the management of diabetic peripheral neuropathy. The results of that study indicated that the administration of ALA for 2–4 weeks was associated with more favorable outcomes for NCS and neuropathic symptoms [12]. However, the effects of orally administered ALA-alone treatments are still unclear.

Pharmacokinetically, ALA has an oral bioavailability of approximately 30% because of its short blood half-life, high presystemic elimination, and hepatic first-pass effect [36]. This phenomenon likely explains the previously uncertain effects of orally administered ALA. Given the emergence of novel technologies and new findings, we included only RCTs that focused on orally administered ALA, with the aim of identifying divergent findings relative to those reported in 2012. Although the RCTs included in the present meta-analysis differed in terms of their intervention period and prescription dose, this meta-analysis revealed that the oral administration of ALA produced significantly more favorable results for TSS, NIS, NDS, and the global satisfaction score. The optimal dose of oral-form ALA is not established yet. Though ALA administration produced favorable effects (600–1800 mg/day) that exhibited dose-related trends on TSS and global satisfaction, the trend is not shown on NIS and NDS.

Because of the diverse nature of diabetic neuropathy, a gold standard for objectively assessing diabetic neuropathy is yet to be established. Existing tests, such as monofilament and sensory testing (pinprick, vibration, and temperature), are operator-dependent, and no standardized method for recording the results in subsequent testing has been developed. NCS and electromyography are unsuitable for evaluating the hypersensitivity of nerves and small-fiber neuropathy. SUDOSCAN and quantitative sensory testing (QST) are effective tools for diagnosing diabetic neuropathy but also exhibit several limitations. QST requires patient cooperation, and its result may be influenced by language barriers, cognitive impairment, and anxiety [37]. SUDOSCAN is used to assess sudomotor function, which may be abnormal in patients with diabetic autonomic neuropathy. Skin biopsy is also a novel method for diagnosing peripheral neuropathy on the basis of intraepidermal nerve fiber density (IENFD); however, it is impractical for monitoring symptom progression or assessing treatment efficacy in clinical settings [38]. In the past two decades, corneal confocal microscopy (CCM) has been extensively studied as a biomarker of small-fiber neuropathy and autonomic neuropathy [39]. There are several assessment parameters with CCM, such as corneal nerve fiber density (CNFD), corneal nerve branch density (CNBD), corneal nerve fiber length (CNFL), inferior whorl length, and tortuosity. Decreased CNFD and CNFL can diagnose DSPN early [40,41]. Not only for diagnoses, CCM also plays a role in predicting the outcome. A rapid loss of CNFL is associated with significant large-fiber impairment at follow up [42]. However, there is no RCT using CCM as a treatment outcome regarding ALA yet. CCM is a noninvasive tool and can serve in an objective correlation to small-fiber outcome in a future DPSN study [39,43]. Thus, the results of most studies that used clinical tools for assessment and follow up may have patient selection bias.

During the process of searching for and collecting studies, we observed the lack of a standardized or appropriate method for measuring and assessing the severity of a patient’s symptoms. Among the studies included in our meta-analysis, the most frequently reported symptom score was the TSS, which considers the common symptoms of DSPN, such as numbness, prickling sensation, burning sensation, and pain. The TSS is fully based on subjective reports, which may introduce bias if a study is not a placebo-controlled study. Thus, several scoring systems have been designed to assess both the symptoms and signs of diabetic neuropathy, especially those pertaining to DSPN. The NDS considers vibration sensation and temperature sensation (both assessed using a tuning fork), pinprick sensation, and ankle reflex. The NIS is a composite score that reflects the muscle weakness, reflex loss, touch pressure, vibration, joint position, joint motion, and pinprick sensation pertaining to the index finger and great toe on both sides of the body. However, these tools are not widely used (only two of the included studies used them) to assess treatment outcomes.

Given our meta-analysis results and the mechanisms of ALA, it is logical to infer ALA has favorable effects on TSS, NDS, NIS, and global satisfaction, but not on VPT and NCS. The TSS, NDS, and NIS involve the scores for small-fiber sensory symptoms. Those positive symptoms are a source of annoyance for most patients, resulting in the levels of global satisfaction being higher in ALA groups than in non-ALA groups. Regarding VPT and NCS, VPT assesses large-fiber damage, whereas NCS is poorly correlated with clinical symptoms involving small fibers.

At present, diabetic neuropathy is not curable and can only be managed by slowing its progression, relieving the pain that it causes, and managing its complications. To delay the progression of diabetic neuropathy, improving glycemic control and implementing lifestyle modifications are recommended [44,45]. To relieve the pain caused by this condition, several pharmacological treatments, including gabapentioids (gabapentin, pregabalin, and mirogabalin), serotonin–norepinephrine reuptake inhibitors (duloxetine, desvenlafaxine), tricyclic antidepressants (amitriptyline), and a sodium channel antagonist (oxcarbazepine, lamotrigine, lacosamide, and valproic acid), are recommended in the Oral and Topical Treatment of Painful Diabetic Polyneuropathy: Practice Guideline Update Summary published in 2022 [46]. For the management of complications, diabetic foot care, chronic ulcer management, and cardiovascular risk factor assessment are key steps that should be undertaken [1].

The present meta-analysis has four limitations, such as the small number of studies included, the lack of unified outcomes for assessing DSPN, and the small sizes of the samples. The heterogeneity of the studies could also have introduced bias into our results. Therefore, further research is necessary to determine the optimal duration of treatment.

## 5. Conclusions

Treatment with ALA had favorable effects on sensory symptoms, but not on muscle power, VPT, or nerve conduction. Moreover, ALA provided symptom relief with a dose-dependent response relative to the placebo for TSS and global satisfaction. Thus, nutritional supplementation for diabetic complications may be a preventive strategy in diabetic care. Additional large-scaled research should be carried out to assess the efficacy of ALA on patients with DSPN.

## Figures and Tables

**Figure 1 nutrients-15-03634-f001:**
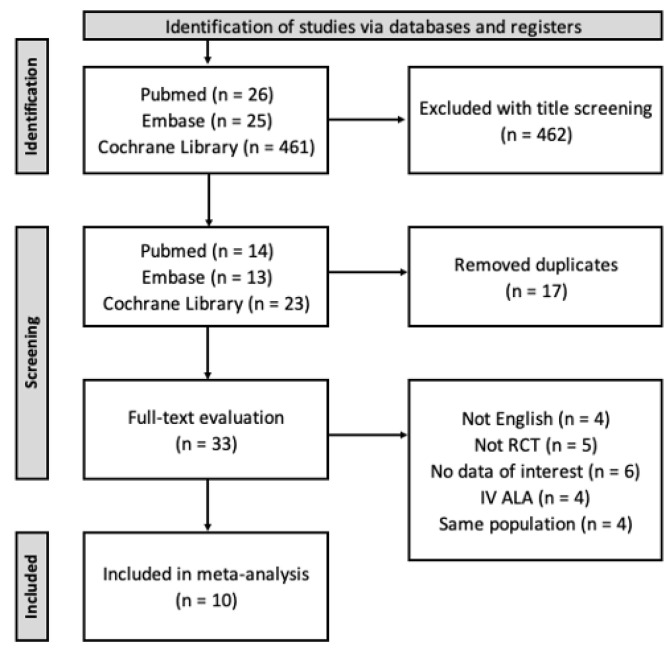
The flow diagram of study selection for the meta-analysis.

**Figure 2 nutrients-15-03634-f002:**
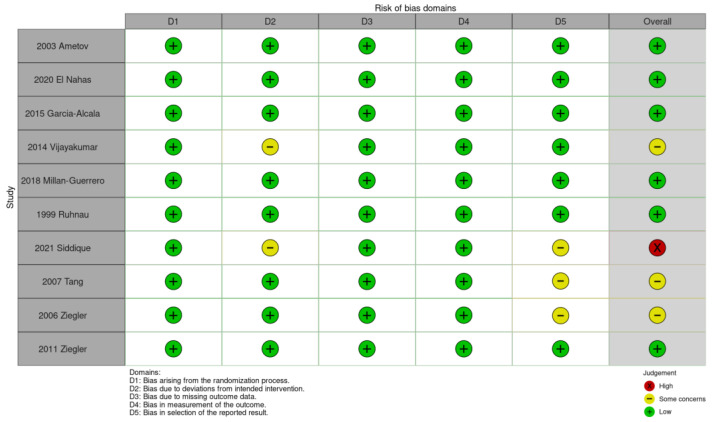
Traffic light plot from the risk-of-bias analysis. Ametov 2003 [14], El Nahas 2020 [19], Garcia-Alcala 2015 [18], Vijayakumar 2014 [25], Millan-Guerrero 2018 [26], Ruhnau 1999 [24], Siddique 2021 [27], Tang 2007 [28], Ziegler 2006 [29], Ziegler 2011 [20].

**Figure 3 nutrients-15-03634-f003:**
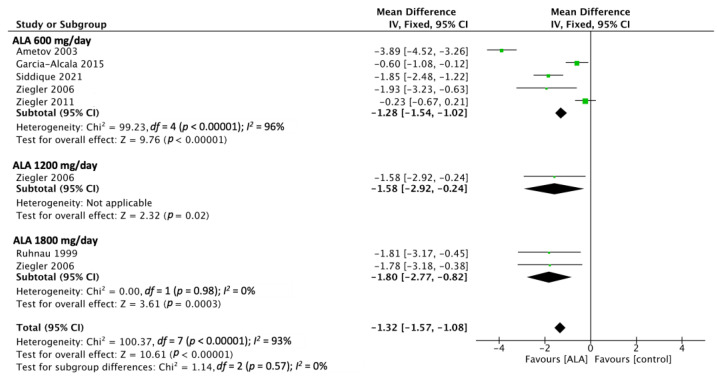
TSS outcome analyzed from six RCTs showed favorable effects on ALA treatment with dose-related trends. Ametov 2003 [14], Garcia-Alcala 2015 [18], Siddique 2021 [27], Ziegler 2006 [29], Ziegler 2011 [20], Ruhnau 1999 [24].

**Figure 4 nutrients-15-03634-f004:**
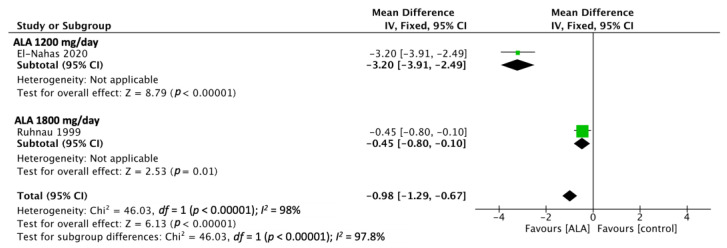
NDS outcomes analyzed from two RCTs showed a favorable effect on ALA treatment. El Nahas 2020 [19], Ruhnau 1999 [24].

**Figure 5 nutrients-15-03634-f005:**
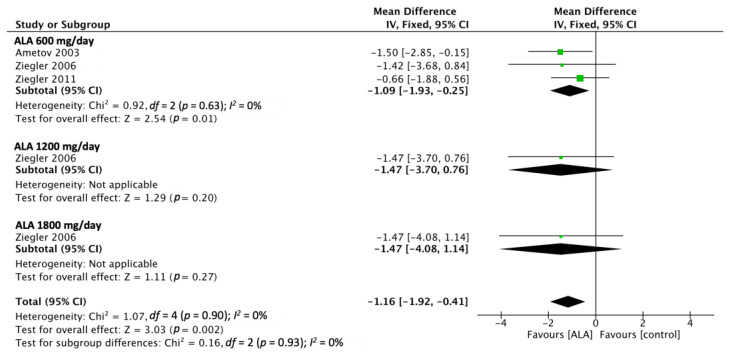
NIS outcomes analyzed from three RCTs showed a favorable effect on ALA treatment. Ametov 2003 [14], Ziegler 2006 [29], Ziegler 2011 [20].

**Figure 6 nutrients-15-03634-f006:**
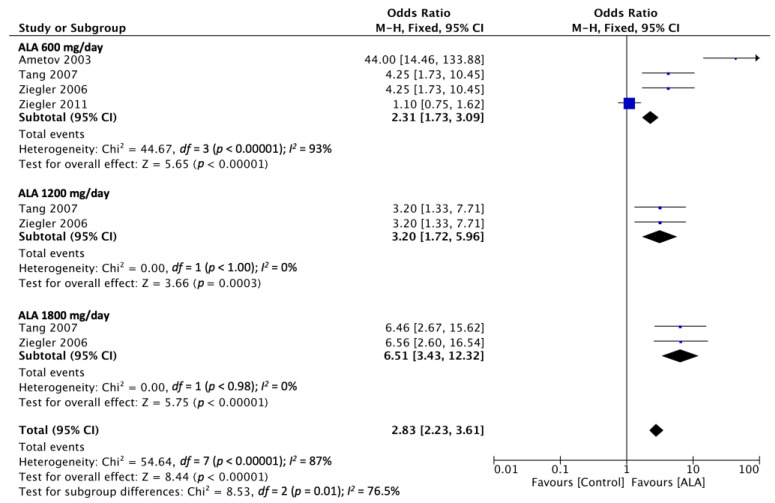
Global satisfaction of ALA at 600 mg/day, 1200 mg/day, and 1800 mg/day. Abbreviation: CI: confidence interval. Ametov 2003 [14], Tang 2007 [28], Ziegler 2006 [29], Ziegler 2011 [20].

**Table 1 nutrients-15-03634-t001:** TSS scoring.

Frequency/Severity	Absent	Mild	Moderate	Severe
Occasionally	0	1.00	2.00	3.00
Often	0	1.33	2.33	3.33
Continuously	0	1.66	2.66	3.66

The TSS is the summation of four sensory symptoms, which ranges from 0 to 14.64. This score was the primary outcome.

**Table 2 nutrients-15-03634-t002:** NDS scoring.

Exam	Score
Vibration (128-tuning fork)	0 = present, 1 = reduce/absent
Temperature (cold-tuning fork)	0 = present, 1 = reduce/absent
Pinprick	0 = present, 1 = reduce/absent
Ankle reflex	0 = normal, 1 = present with reinforcement, 2 = absent

The NDS ranges from 0 to 10. This outcome was treated as a secondary outcome.

**Table 3 nutrients-15-03634-t003:** Summary table of the studies included in this meta-analysis.

Study	Female, %	Age (Mean, Year)	A1c (Mean, %)	DM Duration (Mean, Year)	ALA/Placebo, *n*/N	Dosage, mg	Length, Week	Measures
2003	68%	56.1 ^1^	- ^2^	14.55	60/60	600	14	TSS, NIS, NIS-LL, global satisfaction, NCS
Ametov [14]								
2020	61%	53.4	8.2	11.2	100/100	1200	24	NDS, VPT, VAS
El Nahas [19]								
2015	67%	58.27	8.68	11.74	16/17	600	20	TSS, VPT
Garcia-Alcala [18]								
2014	25%	55.2	-	12.5	10/10	600	12	NCS
Vijayakumar [25]								
2018	60%	50.89	-	10.13	51/49	1200	4	NCS
Millan-Guerrero [26]								
1999	50%	61.3	7.4	11.5	12/12	1800	3	TSS, NDS
Ruhnau [24]								
2021	49%	46.88	8.45	10.64	55/55	600	24	TSS
Siddique [27]								
2007	60%	57.78	7.7	14	138/43	600, 1200, 1800	5	global satisfaction
Tang [28]								
2006	60%	57.78	7.7	14	138/43	600, 1200, 1800	5	TSS, NSC, NIS, NIS-LL, global satisfaction
Ziegler [29]								
2011	67%	53.6	8.85	13.4	230/224	600	104	TSS, NIS, NIS-LL, VPT, NCS
Ziegler [20]								

This table shows demographic data, numbers of patients, and included outcomes of each study. Abbreviations: yr: year; A1c: glycated hemoglobin; DM: diabetes mellitus; ALA: alpha-lipoic acid; TSS: total symptoms’ score; NIS: neuropathy impaired score; NIS-LL: NIS-lower limb; NDS: neurological disability score; VAS: visual analog scale of pain; VPT: vibration perception threshold; NCS: nerve conduction study. ^1^ First number presents mean; ^2^ The missing data are not mentioned in the study reports.

## Data Availability

Data sharing not applicable. No new data were created or analyzed in this study.

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
