# Peer review of "Effects of Oral Alpha-Lipoic Acid Treatment on Diabetic Polyneuropathy: A Meta-Analysis and Systematic Review"

_nutrients, 2023, doi:10.3390/nu15163634_

Round 1
Reviewer 1 Report
Given the current lack of FDA approved pathogenetic therapies for diabetic neuropathy this is an important systematic review assessing the outcomes following ‘oral’ administration of ALA. Previous metanalysis have focused on iv ALA.
As such the methods are appropriate and the outcomes and conclusions clearly stated.
1. Introduction- It is incorrect to state: ‘In the peripheral nervous system, only a few arterioles penetrate the endoneurium to supply the nerve fibers. When the blood flow cannot compensate the decrease of circulation, the individual’s peripheral nerves would be damaged to ischemia.’ Studies from many years ago show a reduction in endoneurial capillary density and pathology of the vessel wall with reduced oxygen diffusion (Malik RA et al Diabetologia. 1989; 32: 92-102).
2. In the results section the duration of the studies should be stated.
3. In the discussion you state: ‘Given our meta-analysis results and the mechanisms of ALA, it is logical to infer ALA has favorable effects on TSS, NDS, NIS, and global satisfaction, but not on VPT and NCS.’ Why should this be the case? Why should a treatment which is supposed to affect underlying causal pathways of DSPN only improve neuropathic symptoms and clinical neurological evaluation (NDS) and yet not objective measures of nerve function like NCS? Why are small fibres more likely to benefit. Perhaps this argues for the need to undertake more detailed evaluation of small fibres using thermal thresholds, IENFD and indeed corneal confocal microscopy, which has been completely ignored, despite published data showing that early benefit is seen in the corneal nerve fibres (Azmi et al Diabetologia. 2019 Aug;62(8):1478-1487).
4. In the conclusion you state: ‘Treatment with ALA had favorable effects on patients with DSPN who exhibited sensory symptoms (i.e., lancinating pain, burning sensation, prickling sensation, and numbness during sleep), but not on muscle power, nerve conduction, and vibration test results.’ However it is more accurate to state that ‘Treatment with ALA had favorable effects on sensory symptoms, but not on muscle power, VPT or nerve conduction.’
Author Response
Dear Editor and Reviewers,
We appreciated your reviewing and providing constructive suggestions for our article. Reviewer 1 mentioned about the alternation of endoneurial capillary density in diabetic neuropathy. Furthermore, corneal confocal microscopy (CCM) could be a sensitive biomarker to evaluate the therapeutic response to ALA. Reviewer 2 pointed out the dosage effect of ALA treatment. It is our belief that the manuscript is substantially improved after making the suggested edits. We hope the manuscript after revisions could meet your high standards and be accepted by Nutrients. The revision has been developed in consultation with all coauthors, and each author has given approval to the final form of this revision.
Below are the point-by-point responses. All the revisions have been highlighted insides the manuscripts already.
Re: Reviewer 1:
- Introduction- It is incorrect to state: ‘In the peripheral nervous system, only a few arterioles penetrate the endoneurium to supply the nerve fibers. When the blood flow cannot compensate the decrease of circulation, the individual’s peripheral nerves would be damaged to ischemia.’ Studies from many years ago show a reduction in endoneurial capillary density and pathology of the vessel wall with reduced oxygen diffusion (Malik RA et al 1989; 32: 92-102).
- Thank you for your suggestion. We have reviewed the article from Malik RA et al about the reduction of capillary density in endoneurium in DSPN patients. Thus, we revised the context to ‘studies from Malik et al also demonstrated that nerve fibers had ischemic changes secondary to reduce endoneurial capillary density’ in page 2 line 53-54.
- In the results section the duration of the studies should be stated.
- We appreciate the reviewer for your constructive suggestions. We have added the duration of each study in every outcome to make the results clearer. The revisions have been highlighted insides the manuscript, and the lines are in page 8 and 9, lines 282-285, 296-298, 305-308, and 318-321.
- In the discussion you state: ‘Given our meta-analysis results and the mechanisms of ALA, it is logical to infer ALA has favorable effects on TSS, NDS, NIS, and global satisfaction, but not on VPT and NCS.’ Why should this be the case? Why should a treatment which is supposed to affect underlying causal pathways of DSPN only improve neuropathic symptoms and clinical neurological evaluation (NDS) and yet not objective measures of nerve function like NCS? Why are small fibres more likely to benefit. Perhaps this argues for the need to undertake more detailed evaluation of small fibres using thermal thresholds, IENFD and indeed corneal confocal microscopy, which has been completely ignored, despite published data showing that early benefit is seen in the corneal nerve fibres (Azmi et al 2019 Aug;62(8):1478-1487).
- We are appreciating for your kindly reminding us to mention important clinical biomarkers of evaluating small fiber neuropathy. We have added prescription about CCM in page 11 and 12, line 404-413.
- In the conclusion you state: ‘Treatment with ALA had favorable effects on patients with DSPN who exhibited sensory symptoms (i.e., lancinating pain, burning sensation, prickling sensation, and numbness during sleep), but not on muscle power, nerve conduction, and vibration test results.’ However, it is more accurate to state that ‘Treatment with ALA had favorable effects on sensory symptoms, but not on muscle power, VPT or nerve conduction.’
- We appreciate your thoughtful suggestion about making our conclusion much easier to read and understand. We have revised the sentence in conclusions in page 12, line 457-458.
We have revised our manuscript as mentioned above and we hope this article could be accepted by your journal.
Sincerely yours,
Ruey-Yu Hsieh and Jia-Ying Sung

Reviewer 2 Report
Hsieh et al present their findings from the meta-analysis in which they evaluated the effects of oral administered ALA as compared to placebo in patients with diabetic sensorimotor peripheral neuropathy. Some minor comments-
The authors should further elaborate on the wide range of the dosage of ALA administered which varied significantly across studies. Differences in dosage may affect outcomes and should be accounted for.
The authors should further explain the methods for measuring and assessing patient symptoms severity and account for differences during analyses.
Author Response
Dear Editor and Reviewers,
We appreciated your reviewing and providing constructive suggestions for our article. Reviewer 1 mentioned about the alternation of endoneurial capillary density in diabetic neuropathy. Furthermore, corneal confocal microscopy (CCM) could be a sensitive biomarker to evaluate the therapeutic response to ALA. Reviewer 2 pointed out the dosage effect of ALA treatment. It is our belief that the manuscript is substantially improved after making the suggested edits. We hope the manuscript after revisions could meet your high standards and be accepted by Nutrients. The revision has been developed in consultation with all coauthors, and each author has given approval to the final form of this revision.
Below are the point-by-point responses. All the revisions have been highlighted insides the manuscripts already.
- The authors should further elaborate on the wide range of the dosage of ALA administered which varied significantly across studies. Differences in dosage may affect outcomes and should be accounted for.
- We appreciate the reviewer for your constructive suggestions. We have added dosage in each study in results (the revisions are highlight in yellow, and the lines are in page 8 and 9, lines 282-285, 296-298, 305-308, and 318-321.) We also added prescription about optimal dosage in page 11, line 387-390.
- The authors should further explain the methods for measuring and assessing patient symptoms severity and account for differences during analyses.
- We appreciate the reviewer for your insightful comments. In our articles, we did not explain how each study done their VPT in detail. Thus, we have further explained the measuring in page 4, line 172-179, which also highlight in yellow.
We have revised our manuscript as mentioned above and we hope this article could be accepted by your journal.
Sincerely yours
Ruey-Yu Hsieh and Jia-Ying Sung
